# Preparation, Properties and Water Dissolution Behavior of Polyethylene Oxide Mats Prepared by Solution Blow Spinning

**DOI:** 10.3390/polym14071299

**Published:** 2022-03-23

**Authors:** Miguel Ángel Lorente, Gustavo González-Gaitano, Javier González-Benito

**Affiliations:** 1Department of Materials Science and Engineering and Chemical Engineering and Institute ITQMAAB, Universidad Carlos III de Madrid, Avda. Universidad 30, 28911 Leganés, Spain; milorent@ing.uc3m.es; 2Department of Chemistry, Universidad de Navarra, 31080 Pamplona, Spain; gaitano@unav.es

**Keywords:** solution blow spinning, polyethylene oxide, morphology, materials characterization, polymer dissolution

## Abstract

The relationship between processing conditions, structure and morphology are key issues to understanding the final properties of materials. For instance, in the case of polymers to be used as scaffolds in tissue engineering, wound dressings and membranes, morphology tuning is essential to control mechanical and wettability behaviors. In this work, the relationship between the processing conditions of the solution blow spinning process (SBS) used to prepare nonwoven mats of polyethylene oxide (PEO), and the structure and morphology of the resulting materials are studied systematically, to account for the thermal and mechanical behaviors and dissolution in water. After finding the optimal SBS processing conditions (air pressure, feed rate, working distance and polymer concentration), the effect of the solvent composition has been considered. The structure and morphology of the blow spun fibers are studied as well as their thermal, mechanical behaviors and dissolution in water. We demonstrate that the morphology of the fibers (size and porosity) changes with the solvent composition, which is reflected in different thermal and mechanical responses and in the dissolution rates of the materials in water.

## 1. Introduction

Polyethylene oxide, PEO, is a biocompatible, water-soluble polymer used extensively for the release of active agents upon its dissolution in an aqueous environment [1,2,3]. The dissolution rate mainly depends on the molar mass of the polymer but also on its crystallinity and morphology, factors that condition the surface accessible to the solvent in the dissolution process. In turn, these depend on the processing method used to produce the material. This is the case of polymeric fibers, whose size, shape, and entanglement degree can be controlled to produce different release profiles. Polymeric fibers are lately receiving much interest for their applications as scaffolds in tissue engineering [4,5], wound dressings [6] and membrane design [7]. Among the different methods to prepare fibrous materials in the form of films or mats, electrospinning, ES, and solution blow spinning, SBS, are the most frequently used. Fiber diameters ranging from a few microns to tens of nanometers can be produced with both methods, depending on the processing conditions [8,9]. In the case of ES, the polymeric solution is injected through a nozzle containing a capillary. The drop that forms at the tip of the capillary is then drawn by the action of an electrical potential between the nozzle and a substrate (collector) [10,11,12] and transforms into dry fibers as the solvent quickly evaporates. In SBS, the drop formed at the tip of the capillary is drawn by the action of a pressurized gas that passes through a concentric nozzle [13,14,15] and favors the evaporation and orientation of fibers towards the collector, usually a rotating cylinder. Although ES is more widespread, the interest in SBS over the last few years has considerably increased, due to its simplicity of handling, no need for the application of large electric fields and in-situ application. On the other hand, there is another technique named forcespinning that has great interest in preparing polymer nanofibers [16,17], however, like ES, it does not seem feasible when the material is expected to be dispensed on specific sites, such as for instance a wound to be treated. Up to now, it can be considered that there are very few articles published on the preparation of polymeric materials from SBS compared to those published where the materials are prepared by ES. For this reason, taking into account the potentiality of SBS to prepare materials for multiple applications, even more work should be conducted, since there are still many aspects to improve, for instance, control of final morphology, less loss of material, among others.

Preparation of fibrous materials by SBS requires a careful choice of the processing conditions. The parameters that can affect the final morphology [18,19,20] can be grouped as shown in Table 1.

The fine adjustment of these parameters can lead to different morphologies [21,22,23] and, consequently, to different mechanical, thermal and wettability properties. In fact, several works about the use of SBS are simply focused on the study of the influence of processing conditions, represented by the parameters shown in Table 1, on the final morphology of the materials prepared [12]. In addition, if active agents, such as therapeutic molecules, such as antimicrobials or anti-inflammatories are incorporated in the fiber structure, with the aim of providing multifunctionality to the material, they must be compatible with the polymer and soluble in the solvent mixture.

Although there are several very interesting biocompatible and biodegradable biomaterials, including BSA, chitosan, etc. [24,25]. PEO is a very simple and low-cost polymer to start exploring how SBS processing conditions affect final morphology. PEO is a linear hydrophilic polymer that is synthesized from ethylene oxide. It is easy to produce, water soluble, non-toxic, or sensitive to pH or physiological fluids. After contact with water, the PEO is hydrated and swollen, resulting in a layer of hydrogel that regulates the subsequent inlet of water and the corresponding dissolution of the active agents that may be inside the polymer. These special properties of hydration and swelling of the PEO make it a remarkably interesting candidate in the pharmaceutical industry for the manufacture of drug release systems [26].

Optimization of SBS processing conditions is thus of great importance in order to achieve the desired properties, which constitutes the objective of the present work, focused on the preparation of fibrous PEO-based mats. The influence of the processing conditions on the morphology and the thermal and mechanical response has been evaluated in terms of the ability of the polymer to be dissolved in water, an essential aspect for the design of PEO mats as a controlled drug delivery system. In this study, the influence of the solvent used to prepare the PEO solutions to be blow spun was considered. To change the solvent properties, a simple variation on the composition in a mixture of acetone and chloroform was chosen in this work.

## 2. Materials and Methods

### 2.1. Materials

Powder PEO (Mv = 100,000, CAS: 25322-68-3, density = 1.13 g/mL) was provided by Sigma-Aldrich (Sant Louis, MO, USA). Solutions of the polymer were prepared in a mixture of acetone, Ac ≥ 99.9% purity (CAS: 67-64-1, M = 58.08 g/mol, bp = 56 °C, density = 0.791 g/mL) and chloroform, Chl ≥ 99% purity (CAS: 67-66-3, M = 119.38 g/mol, bp = 61 °C, density = 1.48 g/mL). Both solvents were supplied by Sigma-Aldrich (Sant Louis, MO, USA).

### 2.2. Preparation of Materials

According to previous studies [27,28] preliminary working conditions for the SBS process were set (capillary diameter 0.6 mm, working distance, WD = 15 cm; injection rate, Fr = 0.5 mL/min; air pressure, Ap = 2 bar; rotational speed of the collector, RSC = 200 rpm; volume of injected solution 10 mL). The process was carried out at 23 °C and 40% relative humidity. For a given composition of the solvent mixture, PEO concentrations were 6, 8, 10, 12 and 14% wt/v. For a selected concentration of PEO, the solvent compositions considered were 4:6, 5:5, 6:4, 7:3, 8:2, 9:1 and 10:0 *v*/*v* in Chl:Ac, in such a way that the final materials obtained were named as PEO-46, PEO-55, PEO-64, PEO-73; PEO-82 and PEO-100, respectively. PEO concentrations lower than 6% led to very thin mats with no fibrous morphology. Likewise, polymer concentrations higher than 14%, and acetone proportion higher than 40% were not considered either due to poor polymer solubility.

### 2.3. Characterization Methods

Viscosity measurements were performed using a Haake iQ viscometer. The sample was placed between two circular plates at constant temperature (25 °C), and subjected to an oscillatory shear force, in the frequency range 100–900 s^−1^, with an acquisition time of 30 s per measurement (100 points).

The structure of the materials was investigated by attenuated total reflectance Fourier transform infrared spectroscopy, ATR-FTIR, and X-ray diffraction, XRD. A Shimadzu IRAffinity-1S Spectrometer, equipped with a Golden Gate ATR accessory (diamond window) was used to collect the IR spectra (32 scans per interferogram), in the wavelength range 600–4000 cm^−1^. XRD patterns were recorded with an X-ray Powder Diffractometer Bruker ECO D8 Advance (Bruker, Karslruhe, Germany) with a Bragg-Brentano configuration and a Lynxeye XE-T coupled detector, using the Cu-K_a1_, λ = 1.5418 Å. Blow spun samples were placed on an amorphous glass holder and the diffractograms were recorded from 5° and 35° in 2θ at 2 s per step with 0.02° of step size.

The thermal behavior of the materials was studied by differential scanning calorimetry, DSC, using a Mettler Toledo 822e calorimeter under N_2_ atmosphere, applying the following thermal cycles: (i) dynamic heating from 30 to 90 °C at 10 °C/min (materials prepared by SBS) and (ii) dynamic cooling from 90 to 30 °C at 10 °C/min (materials after erasing their processing and thermal histories).

Temperatures of melting, T_m_, and crystallization, T_c_, were obtained from the endothermic and exothermic peaks in the heating and cooling scans, respectively. Fusion, ΔH_m_, and crystallization, ΔH_c_, enthalpies were determined from the areas under the endothermic and exothermic peaks, respectively. The crystalline fraction was calculated for the samples produced by SBS before and after erasing their processing and thermal histories, χ_m_ and χ_c_, using Equations (1) and (2), respectively:(1)$$\chi_{m}=\frac{{\Delta H}_{m}}{{\Delta H}_{\infty }}$$
(2)$$\chi_{c}=\frac{{\Delta H}_{c}}{{\Delta H}_{\infty }}$$
where ∆H_∞_ = 197 J/g [29,30,31] is the melting enthalpy of 100% crystalline PEO.

The morphology of the SBS materials was studied by field emission scanning electron microscopy (FESEM) using a TENEO-FEI microscope. The signal from the secondary electrons in the ETD detector was used to generate the FSEM images, applying an acceleration voltage of 5 kV. Samples were carbon-coated to avoid charge accumulation using a Leica EM ACE200 low-vacuum coater. Images were acquired using different tilt angles in order to perform a more complete morphological analysis, which was carried out with ImageJ V.1.52a (National Institute of Mental Health, Bethesda, MD, USA).

The specimens of the blow spun materials were detached from the collector considering its direction of rotation (the longest side of the specimen was parallel to the rotation direction). The length and width of the specimens were measured with a caliper (±0.01 mm accuracy), and the thickness with a Digimatic micrometer (Mitutoyo Corporation, Barcelona, Spain), ±1 μm accuracy. At least six specimens per sample were considered, and the dimensions were the average of five measurements.

The mechanical behavior of the materials was studied using a universal testing machine Microtest TPF-1D at a crosshead speed of 1 mm/min and a load cell of 5 kp. Hard rubber pieces were located on the inner side of the clamps to avoid the specimen sliding during the tests. Six well-dimensioned specimens per material were tested. From the analysis of the resulting stress-strain curves, the following parameters were obtained as averaged values: Young’s modulus, E, from the slope of the stress-strain curve; the tensile stress, σ_r_, representing the maximum stress value at which the specimen breaks, determined as the point at which the maximum stress is reduced by 5% [32]; the yield stress, σ_y_, determined as the stress at which plastic deformation starts (stress-strain plot gives up being linear) [32]; and maximum elongation at failure.

For monitoring the dissolution process of the materials, an optical method was proposed, consisting in recording a video (Optika Microscope 4083.13 camera, exposure time of 39 milliseconds) from the output signal of an optical microscope (Olympus SZ-CTV with a light source Olympus Highlight 2100 and an SZ-40 objective with 40× magnification). The quantification of the fraction of dissolved material was carried out from the loss of transparency of the films by image analysis using video-image file converter software, Prism (V.7.14). A MatLab (R2018b) script was used to transform the raw frames images from the video to 8-bit greyscale and then to black and white, using an intensity threshold of 0.7 for each pixel (1 if the intensity is higher than 0.7, and 0 otherwise). The fraction of pixels with an intensity equal to 1 is taken as the fraction of specimen dissolved, which is then plotted versus time.

## 3. Results and Discussion

Processability by SBS is highly dependent on the viscosity of the solution [33,34]. High viscosities can produce nozzle obstruction or overpressure, while low viscous solutions, mainly associated with low polymer concentrations, usually lead to poor yields of material collected. Additionally, when using solvents of relatively high boiling point, the remaining solvent can be found in the blow-spun material, which favors spreading on the collector and formation of films instead of fibers. Therefore, a preliminary viscosity study using different empirical and semi empirical methods has been carried out in order to optimize the solvent composition for a given PEO concentration.

Viscosity as a function of the polymer concentration for different solvent mixtures is plotted in Figure 1. Different color bars correspond to each composition used (volume ratio) of solvent mixture Chl:Ac. The Huggins-Martins method [35,36] provided in all cases the best fits. Regardless of the solvent composition, there is a general trend where viscosity increases with the concentration of PEO and as expected, the higher the proportion of chloroform the higher the viscosity at a particular concentration of PEO, given the higher viscosity of chloroform (0.542 cP at 25 °C versus 0.316 cP for acetone at 25 °C). At concentrations above 10%, the SBS process became unstable due to overpressure in the nozzle, causing heterogeneous solution ejection. Therefore, the polymer concentration was set at 10% wt in all the SBS experiments since it was the highest concentration that produced a stable jet of solution.

The structure of the prepared materials was tested by FTIR-ATR and XRD. The infrared spectra of the blow spun materials are shown in Figure 2. The characteristic peaks of the solvents used (acetone and chloroform) are not observed in the spectra, for example, the absorption band due to the C-Cl stretching in chloroform at 665 cm^−1^ or the C=O stretching band in acetone at 1710 cm^−1^, which reinforces the idea of the complete evaporation of the solvent during the SBS process.

The band assignation of the PEO was conducted by Yosihara et al. [37]. The methylene C-H stretching appears in the range of 2675 cm^−1^ to 3010 cm^−1^. The bands at 1146 cm^−1^, 1095 cm^−1^ and 1060 cm^−1^ are assigned to a triplet stretching of ether group –C–O–C– [38,39,40,41] and can be used to study structural variations in the PEO since they depend on the chain conformation of semi-crystalline PEO [38,39,40,41]. To this effect, the spectra have been normalized to the absorption at 845 cm^−1^, since that band is usually the least affected by changes in the structure, according to I. Pucic et al. [38]. Although the most intense signal is the one at 1095 cm^−1^, the others follow a different order, since the peak at 1146 cm^−1^ is more intense than the one centered at 1060 cm^−1^ [38]. On the other hand, the presence of a double peak at 1359 cm^−1^ and 1340 cm^−1^ is clear evidence of crystalline PEO [41]. In general, a slight decrease of the relative intensities of the peaks at 1359 cm^−1^, 1146 cm^−1^, 1060 cm^−1^ and 960 cm^−1^, would indicate a lower crystallinity degree with the content in chloroform [41].

Figure 3 shows the baseline corrected and normalized XRD diffractograms of the SBS materials. Characteristics peaks of crystalline PEO [42,43] can be observed at 19.3°, corresponding to the reflections due to the planes (120), and at 23.5°, associated with the planes (112) [23,44,45,46,47]. There are no significant differences among the diffraction patterns but a slight shift to low angles of the peak for the (120) planes and a small variation of the peak half-bandwidths and crystallinity degrees with the proportion of chloroform.

The crystallinity can be estimated semiquantitatively from the amorphous and crystalline contributions of the XRD pattern, deduced from deconvolution by Gaussian functions of the reflections. The fraction of crystalline polymer can be calculated by:(3)$$X_{\mathrm{XRD}}=\frac{\sum_{i}A_{i}}{A_{T}}$$
where *A_i_* represents the area of the deconvoluted peak *i* and *A_T_* is the total area of the diffractogram. Additional parameters can be extracted from the diffractograms. The full width at half maximum, FWHM, of the diffraction peaks, is related to the size of the crystallites, D (Equation (4)), as well as the micro-strain, ε (Equation (5)), due to the presence of defects or the processing conditions.
(4)$$D=\frac{k\lambda }{\beta_{s}\cos\theta }$$
(5)$$\varepsilon =\frac{\beta_{s}}{4\tan\theta }$$
where *k* is a dimensionless shape factor whose value is near to 1; λ is the X-ray wavelength; β_s_ is the FWHM in radians after subtracting the instrumental line broadening (considered negligible with respect to FWHM of a peak in a polymer) and θ is the Bragg angle in radians.

The results of these calculations are shown in Table 2. It can be seen how, as the proportion of chloroform increases, the fraction of the crystalline phase slightly decreases. Besides, although among the solvent mixtures there is not a clear correlation between viscosity and D or ε, when pure chloroform is used to dissolve the polymer, the lowest micro-strain and highest crystallite size and crystallinity fraction are obtained. This result can be ascribed to the higher viscosity of the polymer solution. It is reasonable to think that higher viscosity makes it more difficult to drag the solution and the subsequent polymer chain preorientation. This would imply fewer constrictions for the polymer to crystallize leading to a more relaxed crystalline structure, with larger crystallite sizes and a higher degree of crystallinity.

The thermal behavior of the SBS materials was studied by differential scanning calorimetry, DSC (Figure 4). In the case of the as-received commercial sample, a single homogeneous endothermic peak at 68.9 °C is observed, corresponding to the melting point of PEO reported by other authors [48] and by the supplier. On the other hand, highly heterogeneous endothermic transitions are observed for the blow spun polymer, occurring in a wider range of temperatures and even showing distinct overlapped peaks. This peculiar melting behavior of PEO could be ascribed to an instantaneous fractionation process according to the molar mass. Yan Kuo et al. showed that melting and crystallization points of PEO are highly dependent on its molar mass [49]. In the SBS process, solvent evaporation enriches the vapor phase in the most volatile component (acetone, in this case). Since PEO is less soluble in this solvent, higher proportions of acetone will induce the precipitation of polymer, which begins with the longer PEO chains. In other words, as the evaporation proceeds, fibers will form from polymers of different molar masses, from highest to lowest, in a sort of “time of flight” phase separation, which, leads to a heterogeneous thermal behavior, as detected in our DSC experiments. In general, when SBS PEO samples are prepared from chloroform proportions higher than 70%, a narrower endothermic transition is observed, although it is highly heterogeneous (one peak and two shoulders). These results suggest that the SBS process causes a very heterogeneous crystallization (crystallites and spherulites of different sizes and densities) that strongly depends on the nature of the solvent, tending to be more homogeneous the higher the proportion of chloroform in the mixture is. According to this reasoning, when the proportion of chloroform is high enough, all fractions of PEO can remain together, since chloroform is a better solvent of PEO than acetone, leading to a more homogeneous molar mass distribution in the fibers.

Crystallinity degrees, χ_m_ and χ_c_, obtained from DSC thermograms are collected in Table 3. As can be seen, there are no big differences between the materials prepared with values of crystallinity degree in the range 66–71%, coincident with the range of crystallinities obtained by the XRD results (Table 1), however, no correlations with SBS conditions can be extracted.

It is worth mentioning that, after erasing the processing and thermal history of the samples, one would expect very similar DSC thermograms in the cooling scan, and yet the DSC profiles strongly depend on the conditions of sample preparation. This only can be explained if the thermal treatment chosen to erase the processing history is not as effective as it should to completely disrupt the solid-state structure with complete diffusion of macromolecules, in order to perfectly mix the different molar masses fractions. In other words, although the interactions that maintain macromolecules in an ordered structure are overcome, the complete macromolecular flow was not achieved, keeping in some way a patterned seed for subsequent crystallization. It is clear, therefore, that to fully erase the thermal and processing history, higher temperatures or longer times would be necessary.

The materials morphology was studied by FSEM using the signal from secondary electrons, SE. Figure 5 shows images at different magnifications (500×; 4000× and 8000×) of the blow spun solutions at different solvent compositions.

A highly homogeneous fibrous morphology can be visualized in all cases, with little presence of material accumulation in the typical form of bead-on-a-string shape or as heterogeneous corpuscles. By using different tilt angles of observation, morphological differences were not observed (Appendix A). In terms of heterogeneity, regions with different relative densities and entanglements of fibers exist, regardless of the solvent composition. The results of the fibers diameter analysis (Appendix A) are shown in Table 4 (average diameter, <D>; and standard deviation, σ). It can be observed that the mean diameters do not depend on either the tilt angle of observation or the sample type. Likewise, the width and shape of the distributions are also very similar (Appendix A).

The porosity of the materials is a key parameter regarding the potential applications of the material. Two procedures have been used to estimate the porosity:

(a)Image analysis. Permits obtaining the mean pore area, the number of pores and air area (Table 5). The results can be extrapolated to a 3D network of fibers assuming a homogeneous porosity throughout the sample.
(6)$$A_{\mathrm{Air}}=\mathrm{mean}\;\mathrm{pore}\;\mathrm{area}\times \mathrm{number}\;\mathrm{of}\;\mathrm{pores}$$
(7)$$\mathrm{Porosity}\;\left(\%\right)=\frac{A_{\mathrm{Air}}}{A_{\mathrm{image}}}100$$(b)Gravimetry. The void space or volume of air, V_Air_, can be determined by weighing, using a correction parameter (K) as the ratio between the apparent density (weight of the specimen, *m*, divided by its volume, V_specimen_), ρ_A_, and the density of the bulk material, ρ_R_ [50,51]:


(8)
$$V_{\mathrm{Air}}=\frac{m\left(1-K^{2}\right)}{\rho_{A}}$$



(9)
$$\mathrm{Porosity}\;\left(\%\right)=\frac{V_{\mathrm{Air}}}{V_{\mathrm{especimen}}}100$$


The porosity values, given in percentage of air, are shown in Table 6. As expected, image analysis yields lower values of porosity since a projection in a plane or 2D images is considered. However, the important thing is that both methods do not show any particular trend with respect to the solvent composition used to blow spun the PEO.

Another important morphological feature that may influence the performance of the material is the preferential orientation of the fibers. This calculation was carried out on the SEM images by using the OrientationJ plugin in ImageJ software, the results were plotted as number distribution (fraction of fiber segments) as a function of the orientation angle, φ (Figure 6). In these plots, the number of peaks in the distribution gives information about the number of preferential orientations for the fibers, while the homogeneity of a particular orientation is deduced from the peak width. As can be seen, all the samples present a wide monomodal distribution, indicating no preferential orientation of the fibers.

The mechanical behavior of the materials was studied by tensile tests. Since these materials are highly porous, it is convenient to estimate the parameters that correspond to the bulk material, in addition to those that can be extracted directly from the tensile curves. In the first case, all factors affecting the mechanical behavior must be considered (fiber size, preferential orientation, porosity and crystallinity) but, in the second, consideration of fiber size and porosity would not make any sense.

The mathematical treatment applied to evaluate the porosity contribution from the tensile tests data has been described elsewhere [50]. In Figure 7, values of Young’s modulus are plotted as a function of the solvent composition, with and without porosity correction.

As expected, there is a clear difference between the moduli obtained for the as-prepared samples (red values) and after removing the contribution of the porosity (black values), being in this case around 60% higher, and close to the reported Young’s modulus for PEO [42,43,44,45,46,47,48,49,50,51,52,53,54,55]. On the other hand, Young’s moduli increase when the proportion of chloroform in the solvent is used for SBS regardless of the porosity. Other mechanical parameters were also extracted from the stress-strain plots, maximum stress, σ_max_ (Appendix A), stress at failure, σ_f_ (Appendix A), and yield strength, σ_y_ (Appendix A). All of them show similar tendencies, pointing out that mechanical parameters referring to the material strength increase as the proportion of chloroform in the solvent used for SBS increases. These results indicate that variations of structure and/or morphology are affecting mechanical behavior. As shown above, the porosity is practically constant. Likewise, negligible variations in crystallinity as well as in the preferential orientation of the constituent fibers are observed. However, a parameter that may account for the observed mechanical behavior is the observed heterogeneity of the distributions of molar masses of PEO. When the proportion of chloroform increases, a higher homogeneity in the molar mass distribution is expected and, a more homogeneous distribution of molar masses must improve the transmission of loads because of more effective intermolecular interactions.

Regarding the elastic and plastic behavior, they have been studied from the relative areas obtained from the initial strain value and the yield strain value, A_E_, and from the yield strain and the rupture strain, A_P_, respectively (Figure 8). As can be seen, the higher the chloroform content in the solvent used for SBS, the higher the plastic behavior of the material. This result is typical of semi-crystalline thermoplastic polymers, in which a wider plastic region is found when the interactions between macromolecules are stronger.

The dissolution process of the PEO blow spun materials in water was monitored by optical microscopy. Square specimens were placed between two glass slides and the sample was visualized in an optical microscope (Figure 9) [56,57]. Then, 0.25 mL of deionized water was added in between the glass slides, reaching the specimen by capillarity action. The evolution of the material was then recorded as a function of time using a video camera.

As the dissolution occurs, light reflected due to the opaque solid reduces progressively as the medium becomes transparent. The quantitative monitoring of the process was performed by image analysis from video frames taken at different times (details of the data analysis in Section 2), as shown in Figure 10 and Figure 11.

The fraction of specimen dissolved has been plotted as a function of time in Figure 11. All the dissolution plots show a sigmoidal profile: after an induction time, the dissolution process begins, until the curve levels off, marking the finalization of the dissolution. Under our experimental conditions, the sample is fully dissolved in less than 20 s. Depending on the sample type, the slope of the intermediate region of the plot differs. The PEO dissolution observed in this work is faster than in studies reported [58], since the materials here produced are nonwoven fibrous mats and highly porous. This wide contact surface makes the interaction with the solvent more effective and consequently the dissolution. Following the same reasoning, one would expect higher dissolution rates the thinner the fibers are (and the lower the density of fibers or the higher the porosity). As can be seen in Table 5, the average diameter is similar for all the blow spun material, but with certain differences in terms of porosity parameters (Table 5 and Table 6). For instance, all the samples present similar porosities except PEO-55, which has a slightly lower value. On the other hand, PEO-55 and PEO-82 show a smaller number of pores with a higher mean pore area (Table 5), with these the samples showing different dissolution patterns and a slower dissolution rate.

## 4. Conclusions

The solution blow spinning method, SBS, was used to prepare PEO-based materials in the form of nonwoven mats constituted by submicrometric fibers. By changing the solvent composition used to dissolve the polymer (mixture of acetone and chloroform), different thermal, mechanical and dissolution behaviors of the material were found. From the study of the thermal and mechanical behavior, it is concluded that the composition of the solvent conditions, the evaporation rate, and therefore, the way the solid PEO fibers are formed, in an instantaneous or “time of flight” fractionation of the polymer, is according to the different molar masses. A higher proportion of chloroform favors the homogeneity of the molar mass distribution, which is reflected in better-defined thermal transitions in DSC and higher mechanical strength since the specific interactions occurring between the polymer chains are favored. Finally, the dissolution process of the mats is mainly conditioned by the global morphology of the fibrous material, higher porosity, and small fiber diameter leading to a faster dissolution rate in water, as a consequence of the increased polymer surface available to the solvent.

## Figures and Tables

**Figure 1 polymers-14-01299-f001:**
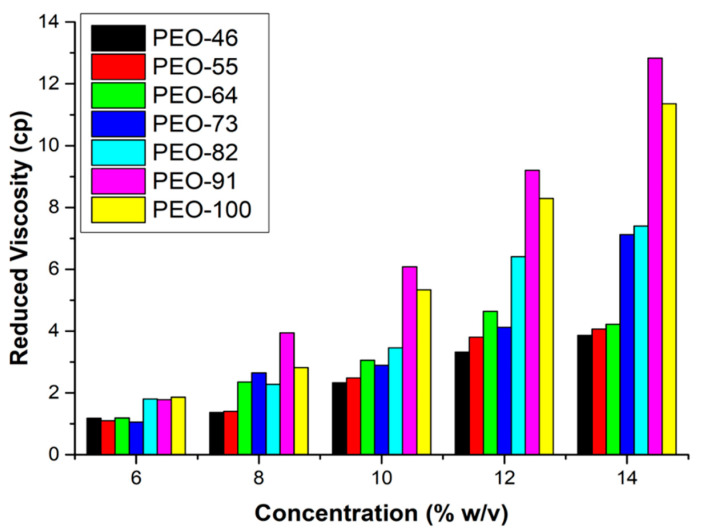
Viscosity of the PEO solutions as a function of concentration and composition of the solvent mixture.

**Figure 2 polymers-14-01299-f002:**
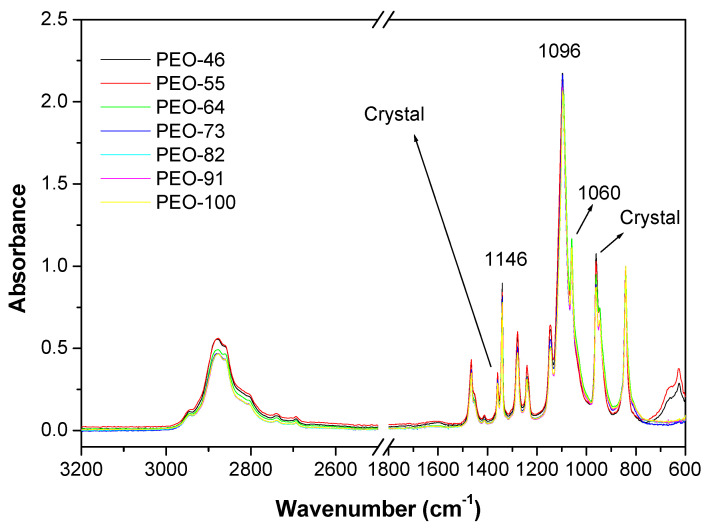
ATR-FTIR spectra of the blow spun PEO-based obtained from 10% wt solutions at different compositions Chl:Ac.

**Figure 3 polymers-14-01299-f003:**
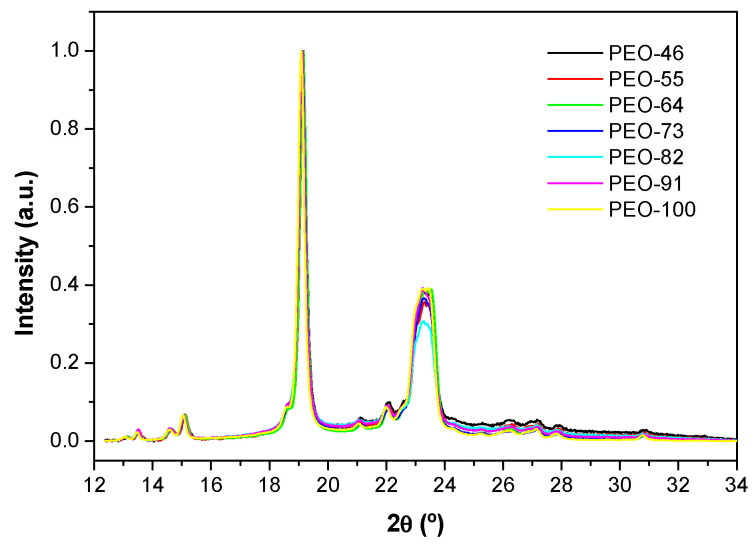
X-ray diffractograms of the blow spun PEO-based obtained from 10% wt solutions at different compositions Chl:Ac.

**Figure 4 polymers-14-01299-f004:**
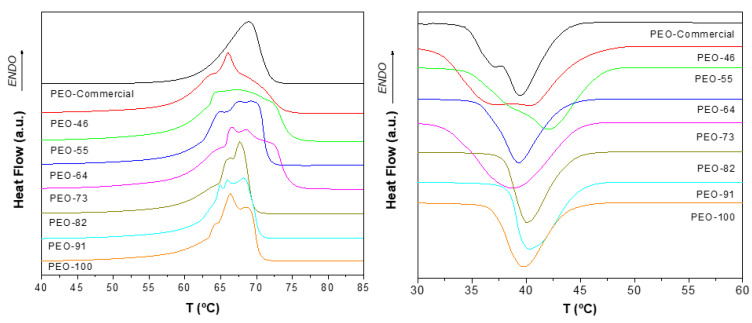
DSC thermograms of blow spun PEO at different compositions Chl:Ac. First heating scan (**left**) and subsequent cooling scan (**right**).

**Figure 5 polymers-14-01299-f005:**
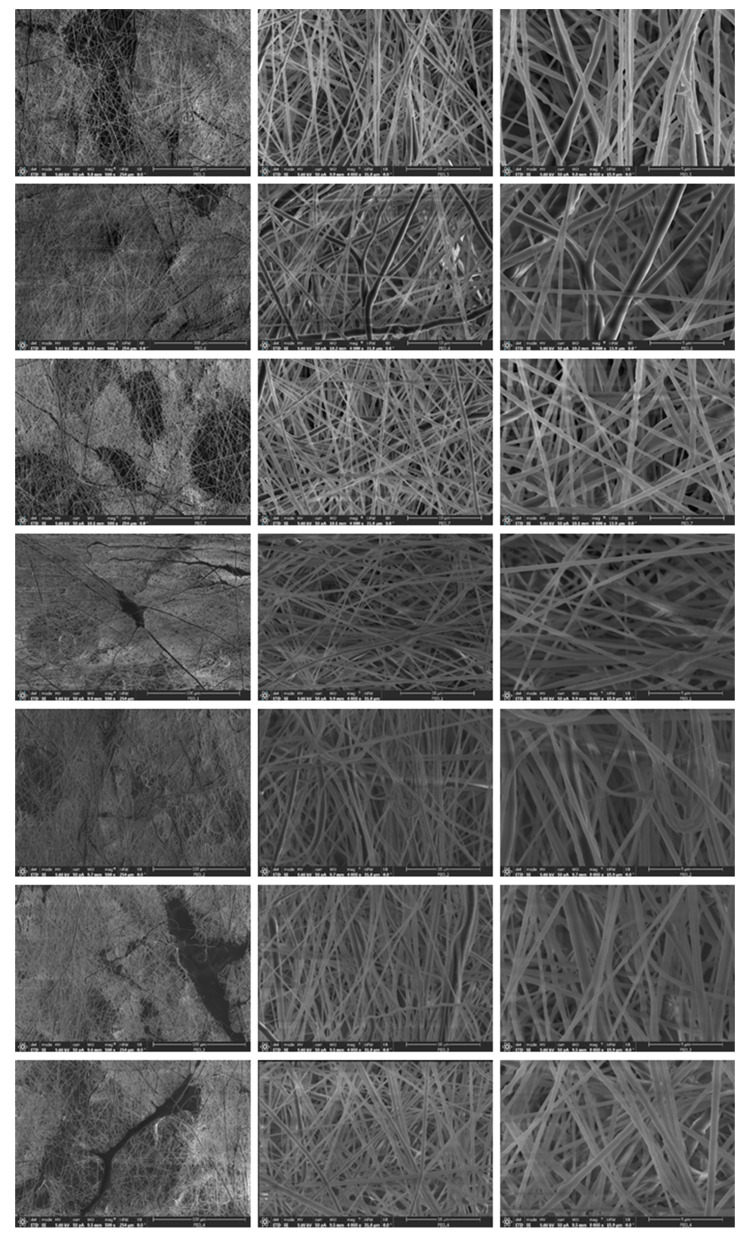
FESEM images of the blow spun fibers at different magnifications. Columns (left to right) represent the magnification, 500×, 4000× and 8000×. Rows (top to bottom) correspond to PEO-46, PEO-55, PEO-64, PEO-73, PEO-82, PEO-91AND PEO-100.

**Figure 6 polymers-14-01299-f006:**
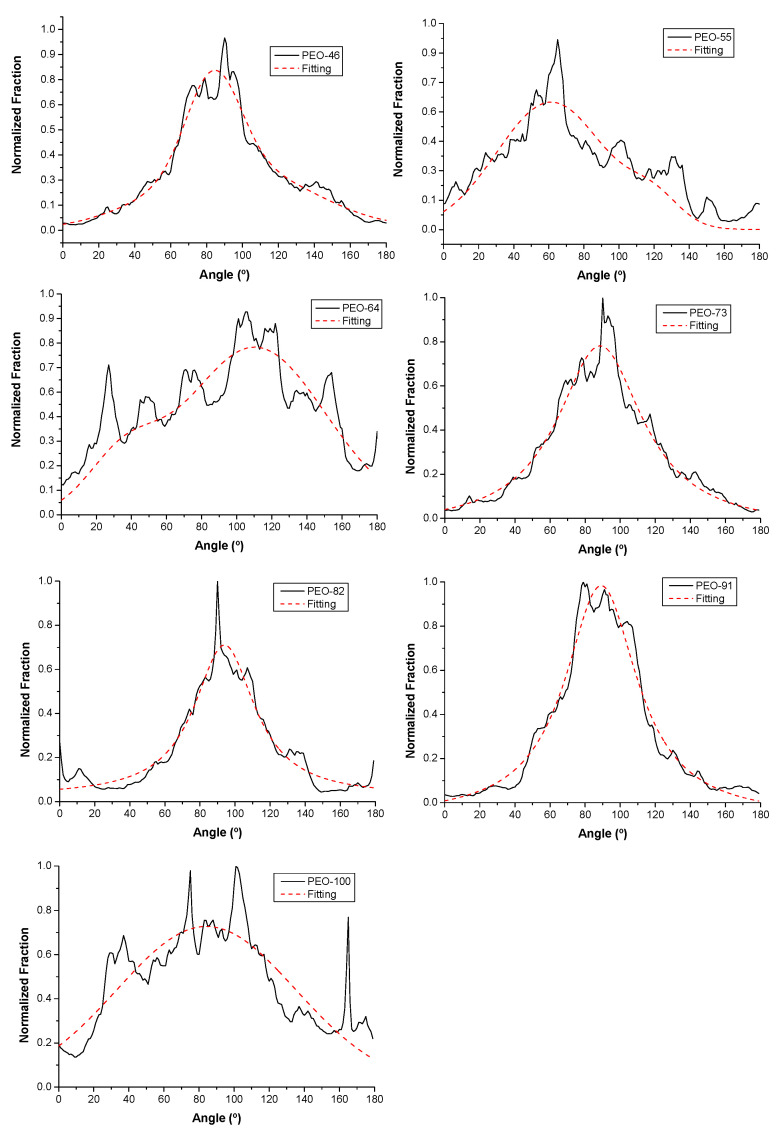
Normalized orientation distributions of blow spun PEO fibers at different compositions Chl:Ac.

**Figure 7 polymers-14-01299-f007:**
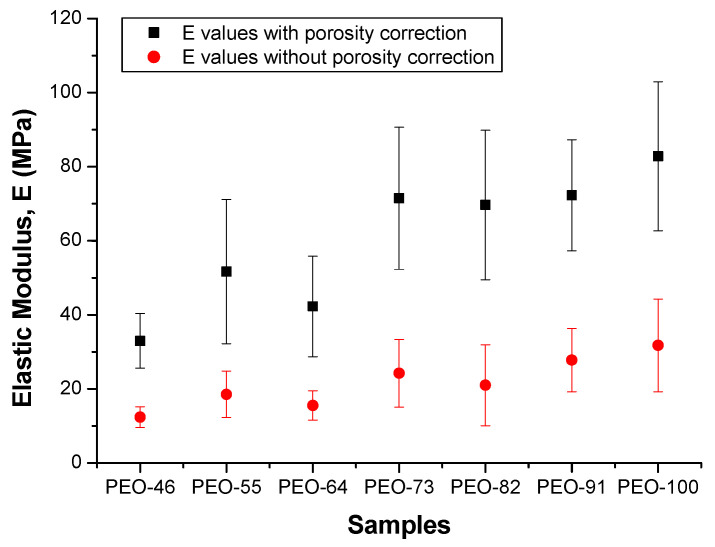
Young’s modulus of the blow spun PEO mats, with and without porosity correction.

**Figure 8 polymers-14-01299-f008:**
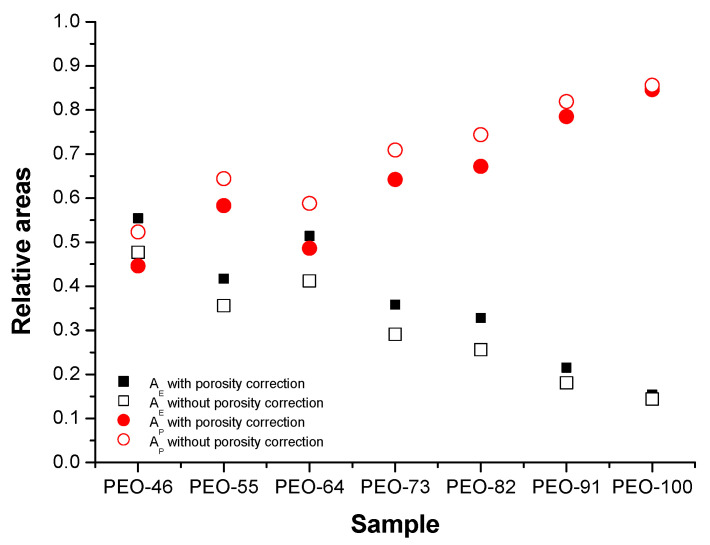
Elastic-plastic behavior of the different PEO-based materials before (circles) and after (squares) considering porosity.

**Figure 9 polymers-14-01299-f009:**
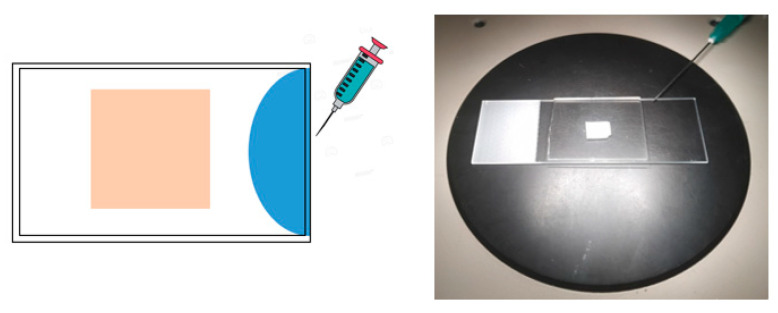
Set up for the dissolution tests.

**Figure 10 polymers-14-01299-f010:**
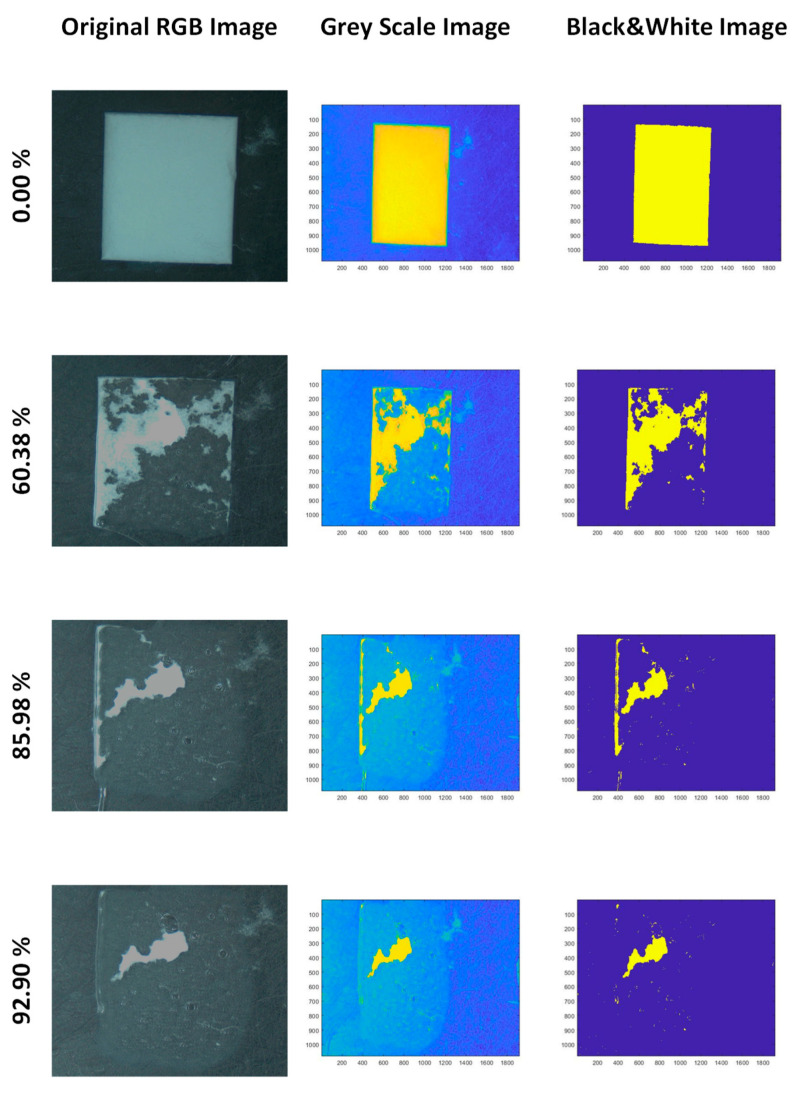
Image analysis to monitor dissolution process of the PEO-based specimens.

**Figure 11 polymers-14-01299-f011:**
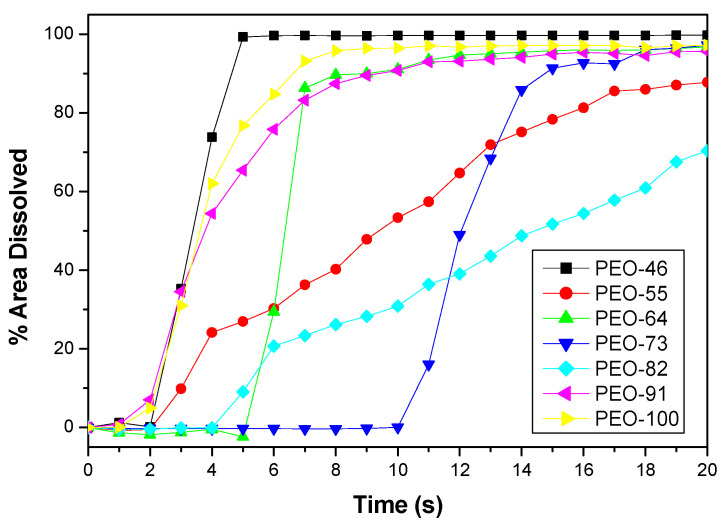
Dissolution profiles of blow spun PEO mats at different solvent compositions Chl:Ac.

**Table 1 polymers-14-01299-t001:** Parameters that affect final morphology of SBS polymer systems.

Parameters Associated to the Solution	Parameters Associated to the Processing	Environment Conditions
Polymer concentration	Working distance, WD (distance from the tip of the capillary to the collector)	Temperature
Type of solvent and composition	Injection or feeding rate, FR (velocity at which the polymer solution is injected)	Humidity
Viscosity	Gas pressure, Ap	
Surface tension	Rotational speed of the collector, RSC	
	Diameter of the capillary	
	Capillary protrusion from the nozzle exit	
Polymer concentration	Working distance, WD (distance from the tip of the capillary to the collector)	Temperature

**Table 2 polymers-14-01299-t002:** Estimated crystalline fraction, crystallite size and micro-strain generated for all materials studied.

	PEO-46	PEO-55	PEO-64	PEO-73	PEO-82	PEO-91	PEO-100
X_XDR_	0.56	0.63	0.74	0.70	0.65	0.70	0.76
D (nm)	34.4	35.2	34.0	33.5	34.0	34.5	35.8
ε × 10^3^	6.74	6.58	6.81	6.93	6.83	6.72	6.50

**Table 3 polymers-14-01299-t003:** Data obtained from DSC curves corresponding to: (a) melting transition and (b) crystallization process.

(a)	PEO 46	PEO 55	PEO 64	PEO 73	PEO 82	PEO 91	PEO 100	PEO Com
T_m_ (°C)	61.9	61.1	61.2	59.0	63.6	55.4	63.1	62.1
∆H_m_ (J/g)	131.6	130.3	126.0	122.4	137.8	145.5	143.1	131.1
χ_m_ (%)	66.8	66.1	64.0	62.2	69.9	73.8	72.6	66.5
**(b)**	**PEO** **46**	**PEO** **55**	**PEO** **64**	**PEO** **73**	**PEO** **82**	**PEO** **91**	**PEO** **100**	**PEO** **Com**
T_c_ (°C)	41.1	46.6	43.1	44.7	43.8	44.9	43.7	43.1
∆H_c_ (J/g)	121.1	126.5	117.8	129.3	123.4	133.4	130.8	88.0
χ_c_ (%)	61.5	64.2	59.8	65.6	62.7	67.7	66.4	44.6

**Table 4 polymers-14-01299-t004:** Morphological parameters of the blow spun fibers extracted from the SEM images analysis.

Sample	Tilt Angle (°)	<D> (nm)	Mean Diameter (nm)	σ (nm)
PEO-46	−5	308	302	240
	0	311		259
	5	302		205
	10	287		187
PEO-55	−5	346	301	302
	0	232		268
	5	316		232
	10	309		193
PEO-64	−5	293	294	237
	0	307		259
	5	289		253
	10	285		235
PEO-73	−5	290	304	180
	0	315		239
	5	324		227
	10	329		246
PEO-82	−5	363	339	290
	0	346		280
	5	295		145
	10	351		187
PEO-91	−5	280	294	225
	0	306		212
	5	274		201
	10	315		238
PEO-100	−5	247	288	169
	0	323		238
	5	319		299
	10	264		187

**Table 5 polymers-14-01299-t005:** Mean pore area estimated from image analysis.

	PEO-46	PEO-55	PEO-64	PEO-73	PEO-82	PEO-91	PEO-100
Mean Pore Area (μm)	0.88	1.28	0.78	0.96	1.21	0.97	0.87

**Table 6 polymers-14-01299-t006:** Estimated porosities deduced from image analysis and gravimetry.

		PEO-46	PEO-55	PEO-64	PEO-73	PEO-82	PEO-91	PEO-100
Porosity (%)	Image Analysis	39	31	37	38	38	38	36
	Gravimetry	62	61	61	68	73	63	64

## Data Availability

Not applicable.

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
