# Peer review of "Preparation, Properties and Water Dissolution Behavior of Polyethylene Oxide Mats Prepared by Solution Blow Spinning"

_polymers, 2022, doi:10.3390/polym14071299_

Round 1
Reviewer 1 Report
The authors have prepared a research article entitled “Preparation, properties and water dissolution behavior of polyethylene oxide mats prepared by solution blow spinning”. The article has some interesting results and the authors have made considerable attention to preparing it. However, some issues need to be clarified before further consideration. Thus, the reviewer recommends this work can be published in Polymers after a major review.
- Rewrite the abstract. Remove characterization techniques such as XRD, FTIR, SEM, DSC, Tensile, etc., and focus directly on the original findings of the current article.
- PEO, SBS, etc., should be placed within Parenthesis
- English should be improved extensively throughout the manuscript.
- A table should be prepared for the following information
‘’a) Parameters associated to the solution, b) Parameters associated to the processing, and c) Environment conditions:’’
- The introduction is very short and should be improved entirely so that the reader can identify the scientific progress of this work.
- There are several biocompatible and biodegradable biomaterials, including BSA, Gelatin, Zein, PCL, PLA, chitosan, UHMWPE, etc. but why did the authors select only PEO? The authors should highlight the favorable characteristics that made PEO a more potent choice of biomaterials for this study. Thus, the authors should emphasize why the PEO are familiar, or favor compared to other biomaterials using the following literature. Moreover, the information on biomaterials (BSA, Gelatin, Zein, PCL, PLA, chitosan, UHMWPE, etc.) should be explored in the introduction with recent references, Thus, the following articles should be quoted in the introduction and other sections.
Chitosan (https://doi.org/10.1016/j.colsurfb.2021.111819), BSA (https://doi.org/10.1016/j.msec.2020.111698),
Gelatin; (https://doi.org/10.3390/ph14040291, https://doi.org/10.1016/j.jmbbm.2020.103696)
Zein (https://doi.org/10.3390/pharmaceutics11120621).
https://doi.org/10.1016/j.msec.2020.110928. https://doi.org/10.1039/D1EN00354B, https://doi.org/10.3390/pharmaceutics12121208
https://doi.org/10.1016/j.jmbbm.2021.104554,
It would be more realistic to cover such kind of research work in the current manuscript. Which will enrich the quality of the current manuscript as well as inquisitiveness to the readers.
- The mechanical properties of PEO-based fibers should be compared with the following forcspun fibers
https://doi.org/10.1039/D1EN00354B
- Even though, there are several techniques are available to produce Nanofibrous scaffolds why did the authors select specifically co-electrospinning? For instance, the authors should compare their solution blow spinning method with Forcespinning® by quoting the following references: https://doi.org/10.1016/j.msec.2020.110928, DOI: 10.1557/mrc.2018.193
- According to the corrections, the conclusions may be modified.
Author Response
Reviewer 1
The authors have prepared a research article entitled “Preparation, properties and water dissolution behavior of polyethylene oxide mats prepared by solution blow spinning”. The article has some interesting results and the authors have made considerable attention to preparing it. However, some issues need to be clarified before further consideration. Thus, the reviewer recommends this work can be published in Polymers after a major review.
- Rewrite the abstract. Remove characterization techniques such as XRD, FTIR, SEM, DSC, Tensile, etc., and focus directly on the original findings of the current article.
Abstract has been changed paying attention to the reviewer’s comments.
- PEO, SBS, etc., should be placed within Parenthesis
Acronyms have been placed in parentheses.
- English should be improved extensively throughout the manuscript.
Following reviewer’s recommendation English has been revised and when necessary improved.
4. A table should be prepared for the following information
4.a) Parameters associated to the solution, b) Parameters associated to the processing, and c) Environment conditions:’
Following the recommendation of the reviewer, a Table (Table 1) gathering the information about factors affecting SBS process has been added. The rest of Tables were conveniently renumbered.
- The introduction is very short and should be improved entirely so that the reader can identify the scientific progress of this work.
The introduction has been slightly enlarged in order to better guide the reader to main aims of the work.
- There are several biocompatible and biodegradable biomaterials, including BSA, Gelatin, Zein, PCL, PLA, chitosan, UHMWPE, etc. but why did the authors select only PEO? The authors should highlight the favorable characteristics that made PEO a more potent choice of biomaterials for this study. Thus, the authors should emphasize why the PEO are familiar, or favor compared to other biomaterials using the following literature. Moreover, the information on biomaterials (BSA, Gelatin, Zein, PCL, PLA, chitosan, UHMWPE, etc.) should be explored in the introduction with recent references, Thus, the following articles should be quoted in the introduction and other sections.
Following the recommendation of the reviewer, favorable characteristics of PEO have been explicitly highlighted in the introduction section and some of the references proposed by the reviewer have been considered in the revised version of the manuscript.
- The mechanical properties of PEO-based fibers should be compared with the following forcspun fibers
https://doi.org/10.1039/D1EN00354B
The article referenced by the above doi corresponds to the study of a membrane system based on a composite of PE filled with carbon nanotubes. We could not find the significance of this comparison if the SBS PEO mechanical results were already compared with mechanical properties of conventional PEO.
- Even though, there are several techniques are available to produce Nanofibrous scaffolds why did the authors select specifically co-electrospinning? For instance, the authors should compare their solution blow spinning method with Forcespinning® by quoting the following references: https://doi.org/10.1016/j.msec.2020.110928, DOI: 10.1557/mrc.2018.193
Following the recommendation of the reviewer, apart from electrospinning and solution blow spinning, forcespinning has been considered in the introduction section as another technique to obtain relatively thin polymeric fibers. One of the references proposed by the reviewer and another (https://doi.org/10.1016/S1369-7021(10)70199-1) were incorporated in the new version of the manuscript.
- According to the corrections, the conclusions may be modified.
In principle, results were not modified consequently the conclusions were not necessary to be modified. Conclusions were just revised in terms of English.
Finally, in the attachment the new version of the manuscript can be found where the reviwer's comments and suggestions were considered.

Reviewer 2 Report
I really enjoyed reading this manuscript. The scientific and linguistic quality is very good and I have no requests for changes or additions. I recommend acceptance without changes.
Author Response
We are very glad to know the reviewer enjoy the article and We would like to appreciate the comments given by him/her since he/she considered the manuscript can be published in Polymers without any change.
Reviewer 3 Report
The manuscript under consideration deals with the processing conditions of solution blow spinning process, SBS, used to prepare nonwoven mats of polyethylene oxide, PEO, and the investigation of structure and morphology of the resulting materials, to account for the thermal and mechanical behaviors and dissolution in water. The work demonstrated significant results and I read this article with interest. Overall, this is a nice well written paper thus can be published .
Author Response
We would like to warmly appreciate the comments given by the reviewer.
Round 2
Reviewer 1 Report
The revised manuscript is acceptable in its present form.